

# The effect of music on the perception of outdoor urban environment

Marek Franěk, Lukáš Režný and Denis Šefara

Faculty of Informatics and Management, University of Hradec Králové, Hradec Králové, Czech Republic

## ABSTRACT

Music may modify the impression of a visual environment. Most studies have explored the effect of music on the perception of various service settings, but the effect of music on the perception of outdoor environments has not yet been adequately explored. Music may make an environment more pleasant and enhance the relaxation effect of outdoor recreational activities. This study investigated the effect of music on the evaluation of urban built and urban natural environments. The participants ($N = 94$) were asked to evaluate five environments in terms of spatio-cognitive and emotional dimensions while listening to music. Two types of music were selected: music with a fast tempo and music with a slow tempo. In contrast with a previous study by *Yamasaki, Yamada & Laukka (2015)*, our experiment revealed that there was only a slight and not significant influence of music on the evaluation of the environment. The effect of music was mediated by the liking of music, but only in the dimensions of *Pleasant* and *Mystery*. The environmental features of the evaluated locations had a stronger effect than music on the evaluation of the environments. Environments with natural elements were perceived as more pleasant, interesting, coherent, and mysterious than urban built environments regardless of the music. It is suggested that the intensity of music may be an important factor in addition to the research methodology, individual variables, and cultural differences.

# INTRODUCTION

There is a large body of studies on auditory and visual interactions. Most studies have explored the effect of music on the perception of service settings (for reviews, see *Garlin & Owen, 2006*; *North, Hargreaves & Krause, 2009*). Another area of research motivated by public health considerations has investigated interactions between perceptions of the outdoor environment and anthropogenic sounds such as traffic noise (for a review, see *Van Renterghem, 2019*). However, the effect of music on the perception of outdoor environments has not yet been adequately explored. Clearly, this problem has practical importance because people often listen to music while relaxing, walking, or running outdoors. Thus, it is useful to know how these musical and visual stimuli interact. Music may modify the impression of surroundings and make an environment more pleasant, interesting, or coherent, thereby enhancing the relaxation effect of various outdoor recreational activities; however, it may potentially have the opposite effect. The present study continues the pioneering investigation of *Yamasaki, Yamada & Laukka (2015)* by

Corresponding author
Marek Franěk, marek.franek@uhk.cz

conducting research in different environmental settings, with different musical stimuli and with ratings of additional perceptual dimensions.

## Everyday music listening

People listen to music in a variety of everyday situations and for many different reasons. Research (e.g., *Greasley & Lamont, 2011*; *Juslin et al., 2008*; *Krause & North, 2017*; *North, Hargreaves & Hargreaves, 2004*) has documented that roughly half of participants' musical experiences occur within the home. People also listen to music frequently while driving (e.g., *Brodsky, 2001*; *Dibben & Williamson, 2007*; *Wiesenthal, Hennessy & Totten, 2003*). Approximately 20% of musical listening occurs in public spaces, such as restaurants and malls (*North, Hargreaves & Krause, 2009*), during exercise (*Hallett & Lamont, 2015*) and on public transport (e.g., *Bull, 2001*; *Lyons et al., 2012*; *Simun, 2009*).

People also listen to music while walking outside (e.g., *Hoffer, 2014*; *Krause & North, 2017*). *Heye & Lamont (2010)* examined some aspects of the effects of listening to music while walking outdoors and showed that music is often used to create an "auditory bubble" (*Bull, 2005*) that changes one's perception of the outdoor environment. The role of an auditory bubble is to help people isolate themselves from unpleasant stimuli from the surrounding environment (*Bull, 2001*). However, in contrast to *Bull (2005)*, *Heye & Lamont (2010)* also observed that their participants often found that the music enhanced their awareness of surroundings rather than distracting them from perceiving their surrounding environment, which shows that the "auditory bubble" is relatively permeable. People listen to music while moving outdoors on portable music devices, smartphones or other equipment via headphones. Listening to music through headphones differs from listening to music through speakers or listening to live music. For instance, it has been shown that listening to speech through headphones helps to fade out surroundings and increases listeners' levels of involvement with the auditory stimulus (*Kallinen & Ravaja, 2007*). Thus, listening to loud music through headphones may result in an attentional shift to the music or even absorption in the music such that the surroundings are not perceived. Therefore, researchers often distinguish between the effects of background and foreground music, particularly in the context of in-store music (e.g., *Yi & Kang, 2019*).

## Interactions between music and the visual environment

There is evidence of the effect of music on the evaluation of the visual environment. Previous studies have predominantly explored the effects of in-store music on customer satisfaction, the perceived atmosphere and customers' consequent willingness to remain in the environment for a longer period of time and spend more money (for reviews, see *Garlin & Owen, 2006*; *North, Hargreaves & Krause, 2009*). However, the effect of music on the perception and evaluation of the surrounding environments during outdoor movement has not been adequately examined. For instance, *Steele et al. (2019)* reported the effects of a "Musikiosk soundscape intervention", which employed an interactive sound system that allowed visitors of a small public park to play music from their own devices in a gazebo over public speakers. It was shown that the park was perceived to be more pleasant during the intervention than it was prior to the intervention; moreover, the perceived calmness

and appropriateness of the soundscape were not affected. However, this study was not a controlled experiment and moreover, the effect of specific types of music on perception of the environment has not been examined. The pioneering study in this field was performed by *Yamasaki, Yamada & Laukka (2015)*, who conducted an experiment in naturalistic conditions in which participants evaluated three outdoor environments and one indoor environment, specifically, a quiet residential area, a busy crossroads, a tranquil park area, and the interior of a train while traveling through the suburbs, while listening to different types of music. Their results showed that highly active music increased the activation ratings of environments that were perceived as inactive without music. In contrast, inactive music decreased the activation ratings of environments that were perceived as highly active without music. Highly positive music increased the positivity ratings of the environments.

A further contribution to this field was also provided by the study by *Bhattacharya & Lindsen (2016)*, who tested whether musical emotion could modulate the judgment of the brightness of a visual pattern. In their investigation participants judged the brightness of a gray square that was presented after an excerpt of emotional music was played. The results showed that happy music made participants judge the subsequently presented gray square to be brighter than the identical gray square presented earlier. In addition, curiously, *Riener et al. (2011)* found that participants who listened to sad music (Mahler's Adagietto) estimated a hill to be steeper than those who listened to happy music (Mozart's Eine Kleine Nachtmusik).

## Effect of intensity and tempo

Clearly, the intensity and tempo of music can have an effect on the perception of the visual environment. Loud music, due to its arousing properties, can increase activation (e.g., *Van Dyck, 2019*). In addition, there is some evidence that the more listeners like a certain type of music, the louder they want to listen to it (*Cullari & Semanchick, 1989*). In accordance with this finding, (*Fucci et al., 1993*) showed that people who liked rock music adjusted the intensity of the music to higher levels than did individuals who disliked rock music.

The tempo of music can modulate the speed of movements in various behavioral domains. For instance, the tempo of music can influence walking speed (e.g., *Franěk, van Noorden & Režný, 2014*; *Leman et al., 2013*; *Styns et al., 2007*) and the speed of various sports activities (e.g., *Karageorghis & Priest, 2012*; *Laukka & Quick, 2013*; *Lane, Davis & Devonport, 2011*). The tempo of music may have an impact on the perception and evaluation of a visual environment. Fast tempo increases arousal as well as ratings of subjective arousal (e.g., *Dillman-Carpentier & Potter, 2007*; *Husain, Thompson & Schellenberg, 2002*). Investigations from the field of consumer research have shown that a fast tempo of in-store music has a mostly positive impact on customers' emotions (*Michel, Baumann & Gayer, 2017*) and might influence visual exploration and the consequent process of consumer decision-making (e.g., *Milliman, 1982*; *Petruzzellis, Chebat & Palumbo, 2015*).

## Environmental preference

When exploring the effect of listening to music on the evaluation of an environment, we must also consider the findings of environmental psychology. Within the field of

environmental psychology and visual aesthetics research, a number of studies, especially in the 1980s, sought to identify the preferred environment and its characteristics. It was repeatedly found that a natural environment containing vegetation was preferred over urban and human-made environments, and in the urban setting, environments that contain some amount of vegetation are preferred (e.g., *Herzog, 1989*; *Kaplan & Kaplan, 1989*; *Purcell & Lamb, 1984*; *Ulrich, 1981*).

*Kaplan & Kaplan (1989)* proposed an information processing approach to explain preferences for specific outdoor environments. They analyzed the information processing of various landscapes and proposed four spatio-cognitive dimensions. (1) *Coherence* is the degree to which environmental elements are related and logically organized. The higher the degree of coherence, the greater the environmental preference. (2) *Legibility* is the extent to which elements allow an observer to understand the environment and its content. The greater the legibility, the greater the preference. (3) *Complexity* involves the number or diversity of elements that the environment contains. The greater the complexity, the greater the preference. (4) *Mystery* is the degree to which the environment contains hidden information. The greater the mystery, the higher the preference. *Herzog (1992)* subsequently proposed several additional spatio-cognitive dimensions (spaciousness, openness, refuge, enclosure, typicality, etc.), but the above mentioned four dimensions are more universal for the description of an environment. These spatio-cognitive dimensions have been widely used in environmental preference research (e.g., *Cassarino & Setti, 2016*; *Herzog, 1992*; *Herzog & Miller, 1998*; *Stamps III, 2004*; *Strumse, 1994*), but they have not yet been employed in investigations of the effects of acoustical stimuli on the perception of visual environments.

Previous research on the effects of music on the evaluation of visual environments was mostly directed by *Mehrabian & Russell*'s theory (*1974*) and predominantly used the affective categories of pleasure (pleasant - unpleasant) and arousal (activating - not activating) to describe the perception of the environment while listening to music. Numerous studies have applied these categories to assess the experience of the environment and its perceived qualities, especially in consumer research (for a review, see *Bitner, 1992*). Thus, the next step in the exploration of auditory-visual interaction is to investigate whether listening to various types of music might influence the perception and evaluation of an outdoor environment in terms of these spatio-cognitive dimensions.

## The current study

The objective of the current study was to replicate the investigation by *Yamasaki, Yamada & Laukka (2015)* and explore the effect of music on perceived emotional and spatio-cognitive dimensions of the environment to be evaluated. In that study, the intensity of music was not controlled, and the sound level was adjusted to an adaptive level by each participant. However, as described above, the intensity of music may be an important factor and should be carefully controlled. Moreover, listening to loud music through headphones may result in an attentional shift to music when the music is in a central position and the surroundings are not perceived (e.g., *Kallinen & Ravaja, 2007*). In this context, *Yamasaki, Yamada & Laukka (2015)* discussed the limitations of their study and emphasized that

the design of their experiment made the musical stimuli more salient to the participants, although music most often takes a less central position in everyday listening situations. They called for varying degrees of musical salience in forthcoming research. Therefore, the intention of the present study was to make music less salient and move it from the foreground to the background. In the current experiment, music was adjusted to be heard at a low level to create an auditory background in which all musical information (melody, tempo, rhythm, timbre) was recognizable, but all attentional resources were not directed to the music. Moreover, in the current experiment, the participants listened to music during the entire experimental session and thus also during the move between evaluated environments, while in the previous study, the participants listened to music only during the proper evaluation of an environment. In this way, we make music less salient and better replicate the realistic conditions of everyday music listening.

Another important variable that may influence the perception of the environment is the tempo of music. Because music at a fast tempo also increases ratings of subjective arousal (e.g., *Dillman-Carpentier & Potter, 2007*; *Husain, Thompson & Schellenberg, 2002*), in the present study, the effect of music at fast and slow tempos is investigated. To increase the activating character of the music, music was selected using the concept of "motivational music" (*Karageorghis et al., 2006*), e.g., music that motivates people to move and supports speed and endurance in various sports activities. Motivational music at a fast tempo motivates movement and is characterized as "Music that gives me a strong urge to move in one way or the other", while the second type of music at a slow tempo, non-motivational music, is described as "Nice music but with no strong urge to move".

Finally, the environmental features of a visual scene are important factors that influence subjective evaluation in terms of the emotional and spatio-cognitive dimensions. As previously discussed, the natural environment is preferred over the urban environment, and the urban environment with natural elements is preferred over the urban built environment without natural elements (e.g., *Herzog, 1989*; *Kaplan & Kaplan, 1989*; *Purcell & Lamb, 1984*; *Ulrich, 1981*).

In summary, the objective of the current study was to explore the effect of fast (motivational) and slow (non-motivational) music on the evaluation of diverse types of outdoor urban environments. Participants evaluated both urban built and urban natural environments while listening to music. The effects of music in terms of the emotional and spatio-cognitive dimensions of the perceived environment were investigated. Based on the results of *Yamasaki, Yamada & Laukka (2015)*, we hypothesized that motivational music may increase the activation rating of an environment, especially in calm environments, while non-motivational music may have the opposite effect. Both types of music may increase pleasure ratings, especially in less attractive environments. Because the current study has an exploratory character, we did not have a specific prediction regarding the effect of music on rating of the environment in terms of the above described spatio-cognitive dimensions. It is unclear whether music can especially influence perceptions of coherence, complexity, and legibility and whether the effect of slow and fast music differs. For instance, non-motivational music in a slow tempo may increase the perceived mystery

of an environment, while motivational music in a fast tempo may increase the perceived openness of an environment.

## METHOD

### Participants

Ninety-four undergraduates participated in the experiment. The sample comprised young adults between the ages of 19 and 23 (Mean = 20.3, SD = 2.17, 58 females). Participants were enrolled in the first, second or third year of various psychology courses. They were students in informatics, financial management and tourism at the University of Hradec Králové.

### Stimulus material

The musical stimuli used in our previous study (*Franěk, van Noorden & Režný, 2014*) were adopted for the present experiment. Before the study (*Franěk, van Noorden & Režný, 2014*), each participant was asked to select and submit two files of different types of music that they liked. The first type of requested music was motivational music (music that motivates one to move), which was characterized as ''music that gives me a strong urge to move in one way or the other'', and the second type of requested music was non-motivational music, which was described as ''nice music but that provides no strong urge to move''. Then, the participants were asked to evaluate the motivational character of the collected musical files, which were made available on a network disk using the Brunel Music Rating Inventory-2 (*Karageorghis et al., 2006*). The inventory consists of the following statements: (1) The rhythm of this music would motivate me during exercise; (2) The style of this music (i.e., rock, dance, jazz, hip-hop, etc.) would motivate me during exercise; (3) The melody (tune) of this music would motivate me during exercise; (4) The tempo (speed) of this music would motivate during exercise; (5) The sound of the instruments used (i.e., guitar, synthesizer, saxophone, etc.) would motivate me during exercise; and (6) The beat of this music would motivate me during exercise. The participants were asked to rate their level of agreement with the statement using a 1-7 scale (strongly disagree - strongly agree). Based on the evaluation using the Brunel Music Rating Inventory-2, the nine musical pieces rated as having the highest motivational character (fast, motivational music) and the nine pieces rated as having the lowest motivational character (slow, non-motivational music) were selected. The scores of the of motivational music pieces ranged from 27.4 to 30.4 points on the Brunel Music Rating Inventory, and the scores of the non-motivational music pieces ranged from 6.6 to 9.4 points.

For the purposes of the present study, we chose two songs from this selection, specifically the song ''One Fine Day'' (The Offspring, album Conspiracy of One, 2000) with a tempo of 187 beats per minute (bpm), which was used as the motivational music (the score of the Brunel Music Rating Inventory-2 was 28.8), and the song ''Madworld'' (Michael Andrews, movie ''Donnie Darko'', 2001) with a tempo of 91 bpm, which was used as the non-motivational music (the score of the Brunel Music Rating Inventory-2 was 7.6). By using *VirtualDJ* software, a long seamless loop was used in both songs to create soundtracks with a duration of approximately 30 min.

The sound levels of the musical tracks were adjusted to be audible as a musical background only. The sound levels of specific musical tracks were adjusted according to the assessment of the team of experimental assistants to have a low level of intensity and were constant during the experiment. The sounds were listened to through lightweight headphones (Genius HS-M200C) connected to a Nokia Lumia 520 phone, on which the musical tracks were recorded.

## Measures

A questionnaire that contained eleven items was used in the experiment (Table 1). Item 1 was related to the spatio-cognitive dimension *Coherence*, item 2 was related to the spatio-cognitive dimension *Legibility*, item 3 was related to the spatio-cognitive dimension *Complexity*, item 4 was related to the spatio-cognitive dimension *Mystery,* and item 5 was related to the spatio-cognitive dimension *Openness*. These items were selected from the study by *Herzog (1992)*. Item 6 was related to the emotional dimension *Energy,* item 7 was related to the emotional dimension *Abandoned*, item 8 was related to the emotional dimension *Pleasant*, and item 9 was related to the emotional dimension *Interesting*. Items 6–9 were selected from the more extensive questionnaire used in the study by *Yamasaki, Yamada & Laukka (2015)*. In contrast with this study, we used only four items from their questionnaire to make the time to complete the questionnaire shorter and facilitate the participants' tasks. When using a long questionnaire, there is a risk that the participants will not pay enough attention to the responses to the individual items, especially when the research is conducted outdoors. Finally, the participants were asked whether they liked the music and whether the music disturbed them. Each item was assessed on a five-point scale (1 = not at all, 5 = completely). The questionnaire was presented to the participants in an electronic form. They read particular items of the questionnaire on their own smartphones and expressed their level of agreement/disagreement on the scales, which contained a link to an online survey.

## Environments

The experiment was conducted in the city of Hradec Králové in the Czech Republic. The city is located in the northeastern part of the Czech Republic and has approximately 100,000 residents. Six outdoor locations were selected (Fig. 1 and Table 2). We chose places on one walking route that were not too far apart and could be walked within roughly 15 min. Greater distance between the locations would require a car ride, which could have some side effects. In this situation, the participants could observe various surroundings during car transportation, and their attention to the task could be distracted.

The initial location was situated in front of the university building. The site was used only for training participants in the evaluation of the environment and using the electronic questionnaire. The obtained data were not further processed. The first environment, where the data were processed, was chosen as an example of an ugly and unorganized urban environment. The second environment was represented by an urban environment with traffic. The third environment was a coherent natural environment with water. The fourth environment represented a coherent natural environment with a high level of spatial

**Table 1  The questionnaire used in the study.**

| | Item | Dimensions |
|---|---|---|
| | | **Spatio–cognitive dimensions** |
| 1. | The individual features of this place are in harmony; they belong together. | *Coherence* |
| 2. | It's easy to understand where I am now, how to get out of here, and where to go next. | *Legibility* |
| 3. | This place contains a large number of various elements. | *Complexity* |
| 4. | I would like to explore this interesting place more. | *Mystery* |
| 5. | This space is rather open. | *Openness* |
| | | **Emotional dimensions** |
| 6. | I feel a strong energy here. | *Energy* |
| 7. | This place seems quite abandoned to me. | *Abandoned* |
| 8. | I feel very comfortable here. | *Pleasant* |
| 9. | It's a pretty interesting place. | *Interesting* |
| 10. | I liked the music. | |
| 11. | The music disturbed me when evaluating the environment. | |

**Table 2  Description of the environments that were evaluated.**

| Environment | Type of environment | Description |
|---|---|---|
| 1 | urban built | A sidewalk with a view of a small parking lot, the backside of a school canteen, waste containers, and a street with a modern building. |
| 2 | urban built | Crossroad; on the left side behind the crossroad was a waterworks building, in the background were residential houses, and on the hill were the towers of a Gothic church. On the right side was a railing of a riverbank, and behind it was the school building. |
| 3 | urban natural | Situated directly above the river, trees visible on both the left and right sides of the river. |
| 4 | urban natural | A meadow. A row of trees at the end of the meadow. |
| 5 | urban natural | Dense oak alley, a meadow at the left side of the alley. |

openness. The fifth environment was a coherent natural environment with a lower level of spatial openness. All environments were familiar to the participants because they were located close to the university buildings.

## Procedure

Before the experiment, the participants signed informed consent and then were instructed. They were informed that their task was to take a route near the university along the Orlice River within a smaller group of people and to evaluate 6 selected sites by means of a questionnaire. They were also informed that they would either listen to music through headphones during the experiment or would take part in the experiment without listening to any music.

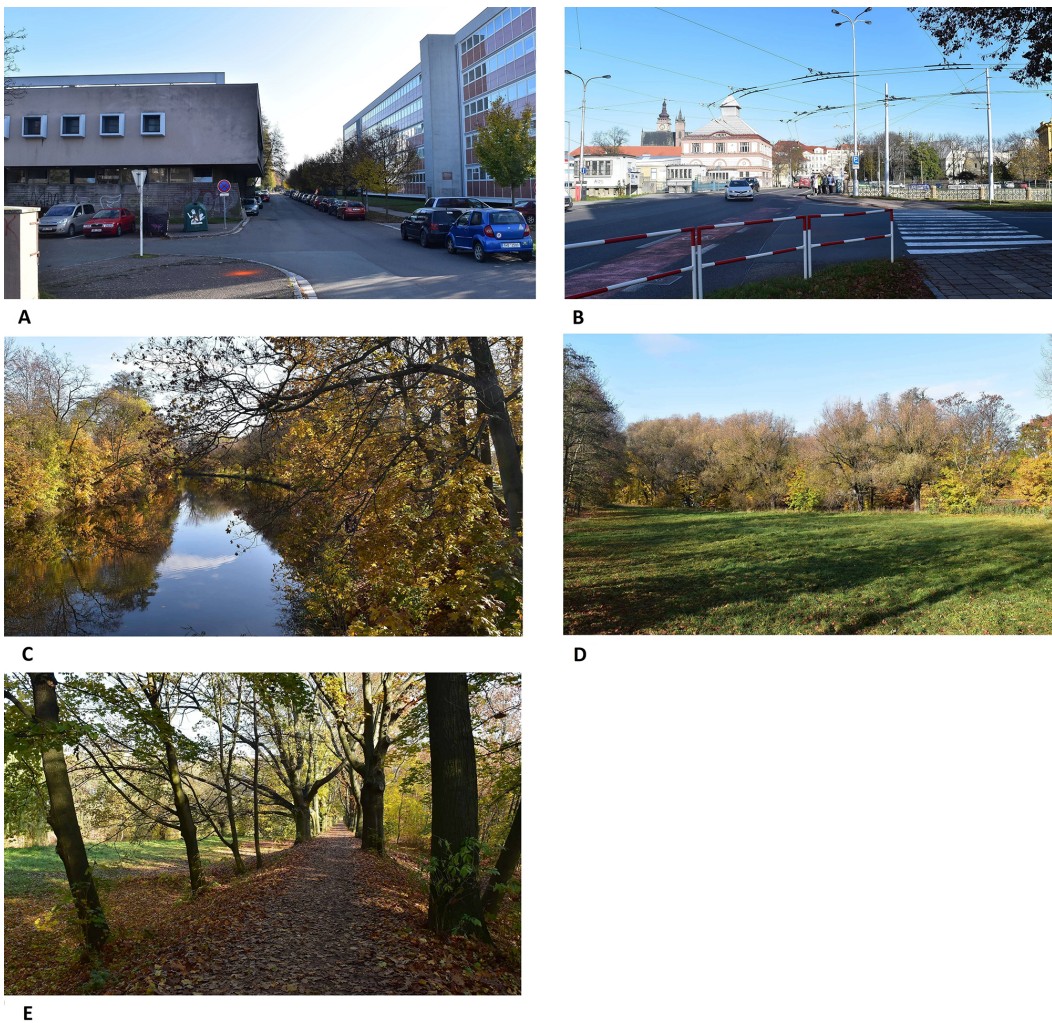

**Figure 1** **The environments that were evaluated** (A) Environment 1, (B) Environment 2, (C) Environment 3, (D) Environment 4, (E) Environment 5.

A between-subjects design was used. Thirty-three participants were randomly assigned to the motivational music condition, twenty-eight participants were assigned to the non-motivational music condition, and thirty-three participants were assigned to the control condition. Because not all preliminarily registered participants came to the experiment (due to illness, etc.), the sample sizes of both music conditions differed.

The participants were asked to find the questionnaire on their smartphones (by using a QR code). The participants in the music conditions also received Nokia phones with recorded music and headphones. The research assistant turned on the music at the initial location. The participants listened to music without any interruption during the entire walk along the route and at the locations where they were asked to evaluate the environment. The participants moved along the route in groups of 2–6 people and were accompanied by a research assistant. The research assistant stopped the participants at the specific location,

showed them the area to observe and asked them to make their records in the questionnaire on their smartphones. The walking distance between specific locations was approximately 2–3 min. To eliminate a possible order effect, the participants moved along the route from environment 1 to environment 5 or from environment 5 to environment 1. The study was conducted in 2019 on three weekdays: November 12, November 13, and November 14. It was cloudy, and the temperature was approximately 8 °C.

## Ethical statement

Ethical approval for the experiments was obtained from the Committee for Research Ethics at the University of Hradec Králové (No. 8/2019). All participants signed a consent declaration in which they declared that they voluntarily participated in the experiment and that they were informed about the experimental procedure. They could decide to stop participating in the research study at any time without explanation. There were no known risks to the participants in this study.

## RESULTS

### Effects of musical condition and environment

The software *Statistica 12* (Stat Soft, Inc.) was used to perform statistical analyses. First, the mean scores and standard deviations for all spatio-cognitive and emotional dimensions were calculated separately for the five evaluated environments (Fig. 2). Next, two-way mixed ANOVAs were conducted separately for all spatio-cognitive and emotional dimensions to assess the effects of the music condition (motivational music, non-motivational music, control condition) and the evaluated environment (environments 1–5) on the mean scores of particular evaluated dimensions. The music condition was chosen as a between-subjects factor, and the environment was chosen as a within-subjects repeat factor. All the ANOVAs revealed strong significant effects of the evaluated environment, while the music condition had no significant effects. Moreover, there were no significant interactions between the condition and the section. The results of all ANOVAs are listed in Table 3.

In contrast to our prediction, music had no significant effect on the evaluation of the observed environments. Although there were some differences between the effects of specific music (Fig. 2), they were small and statistically nonsignificant. However, the results revealed significant differences in the evaluation of the environments, namely, between both urban environments and between urban and natural environments.

The mean scores of *Coherence* were lowest in both urban environments; in the natural environments, 3 and 4 were significantly higher, and the highest score was in environment 5, which was formed by an oak alley. In contrast, the mean *Legibility* score was significantly highest for urban environment 2, formed by a crossroad and sidewalks. Within the natural environments, the significantly lowest score was for environment 4, formed by a meadow. Similarly, the mean scores of *Complexity* were higher for urban environments than for natural environments. The significantly highest score for perceived complexity was in environment 2, formed by a crossroad and sidewalks with a view of the towers in the central area of the city. The significantly lowest score for perceived complexity was for natural environment 4, formed by a meadow. In contrast, the mean scores of *Mystery* were

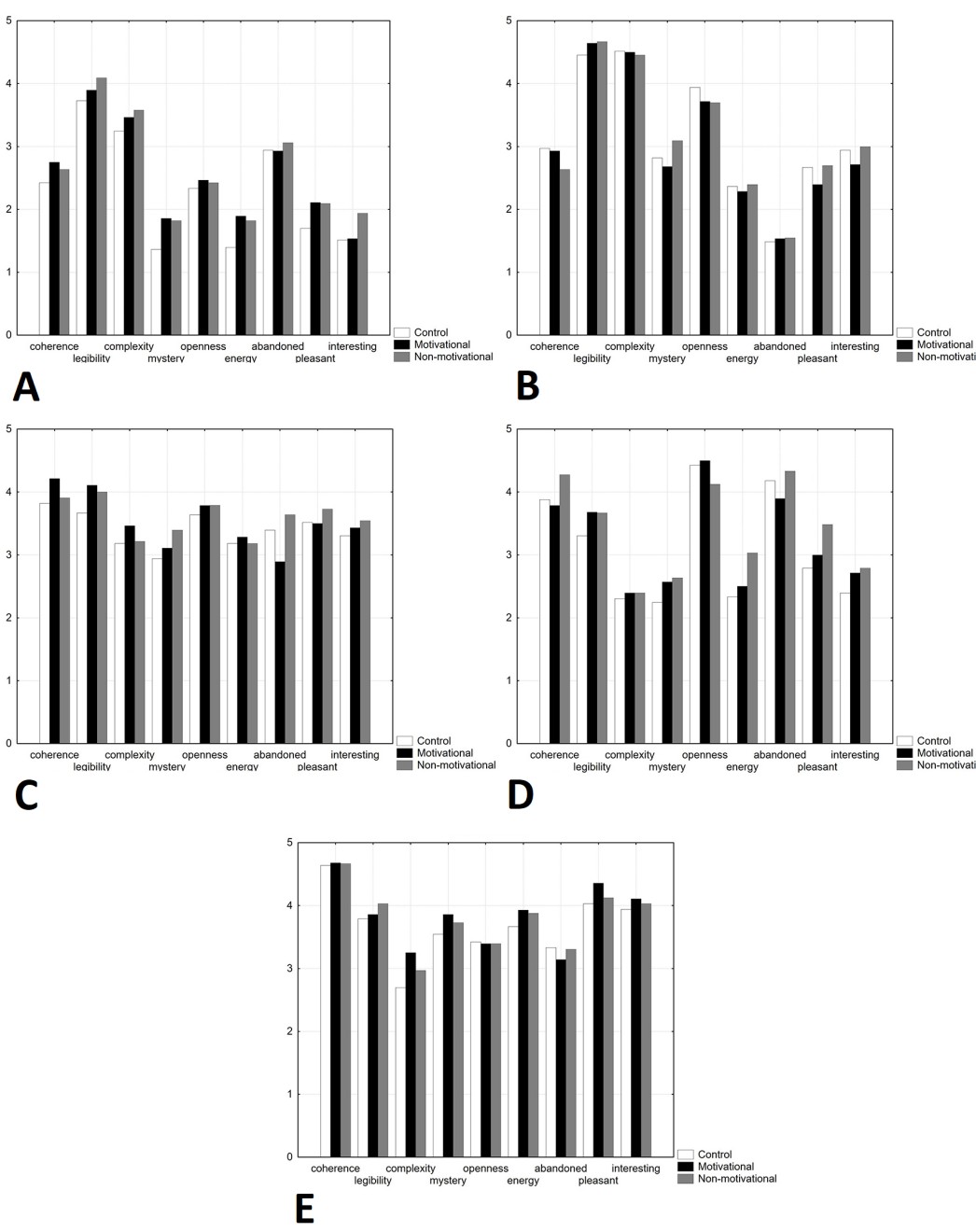

**Figure 2   Mean scores for all spatio-cognitive and emotional dimensions for environments 1–5 separately** The scale ranged from 1 to 5. Empty columns represent the control condition, black columns represent motivational music 1, and gray columns represent non-motivational 2. (A) Environment 1, (B) Environment 2, (C) Environment 3, (D) Environment 4, (E) Environment 5.

**Table 3  Results of the ANOVA for all evaluated spatio-cognitive and emotional dimensions.** The results for a between-subjects factor "Condition" and the within-subjects factor "Environment" are presented in columns. Statistically significant results are indicated with boldface. The results of a Tukey post-hoc test for the variable "Environment" are shown in the last column.

| Item | | Condition | Environment | |
|---|---|---|---|---|
| | *df* | 2, 91 | 4, 364 | |
| 1. Coherence | *F* | 0.458 | **68.400** | Significant differences between the environments except environments 1–2, 3–4 |
| | *p* | 0.634 | **<0.001** | |
| | η2 | 0.010 | **0.432** | |
| 2. Legibility | *F* | 1.179 | **19.083** | Significant differences between the environments except environments 1–3, 1–5, 3–5 |
| | *p* | 0.312 | **<0.001** | |
| | η2 | 0.025 | **0.173** | |
| 3. Complexity | *F* | 0.831 | **62.446** | Significant differences between the environments except environments 1–3, 3–5 |
| | *p* | 0.439 | **<0.001** | |
| | η2 | 0.018 | **0.407** | |
| 4. Mystery | *F* | 2.003 | **51.293** | Significant differences between the s environments except environments 2–3, 2–4 |
| | *p* | 0.141 | **<0.001** | |
| | η2 | 0.042 | **0.360** | |
| 5. Openness | *F* | 0.178 | **47.586** | Significant differences between the environments except environments 2–3, 2–5, 3–5 |
| | *p* | 0.837 | **<0.001** | |
| | η2 | 0.004 | **0.343** | |
| 6. Energy | *F* | 1.367 | **68.618** | Significant differences between the environments except environments 2–4 |
| | *p* | 0.260 | **<0.001** | |
| | η2 | 0.029 | **0.430** | |
| 7. Abandoned | *F* | 2.470 | **75.430** | Significant differences between the environments except environments 1–3, 1–5, 3–5 |
| | *p* | 0.090 | **<0.001** | |
| | η2 | 0.051 | **0.453** | |
| 8. Pleasant | *F* | 1.904 | **88.965** | All differences between the environments are significant |
| | *p* | 0.155 | **<0.001** | |
| | η2 | 0.040 | **0.494** | |
| 9. Interesting | *F* | 1.319 | **84.846** | Significant differences between the environments except environments 2–4 |
| | *p* | 0.273 | **<0.001** | |
| | η2 | 0.028 | **0.483** | |

higher for the natural environments than for the urban environments with the exception of the open natural environment 4. The significantly highest level of perceived mystery was found for natural environment 5, formed by an oak alley, and the significantly lowest level of perceived mystery was found in urban environment 1, formed by parking lots and the backside of a school canteen. In the mean scores of *Openness,* there were no systematic differences between urban and natural environments. The highest levels of

perceived openness were found in natural environment 4, formed by a meadow, natural environment 3, with the view of the river, and urban environment 2, formed by a crossroad with a view of the towers in the central area of the city.

The mean scores of *Energy* were higher in natural environments than in urban environments. The significantly lowest level of perceived energy was in environment 1, formed by parking lots and the backside of a school canteen, while the significantly highest level of perceived energy was, curiously, in environment 5, formed by a calm oak alley. Clearly, the perceived levels of *Abandonment* were higher in natural environments. The significantly highest level of perceived *Abandonment* was in environment 4, formed by a large empty meadow, while the significantly lowest level of perceived *Abandonment* was in environment 2, formed by a relatively busy crossroad. The mean scores of *Pleasant* were higher in natural environments than in urban environments. The significantly highest level of perceived pleasantness was in environment 5, formed by an oak alley, while the significantly lowest level of perceived pleasantness was in environment 1, formed by parking lots and the backside of a school canteen. The mean scores of the dimension of *Interest* were higher in natural environments than in urban environments. The significantly highest level of perceived interest was in environment 5, formed by an oak alley, while the significantly lowest level of perceived interest was in environment 1, formed by parking lots and the backside of a school canteen.

## Effects of music liking

Given that the ANOVAs did not identify any significant effect of music on the evaluation of the environment, we examined whether the evaluation of an environment was mediated by the listener's level of liking of the music. The levels of liking of motivational music (Mean 3.75, SD = 0.93) and non-motivational music (Mean = 3.81, SD = 1.26) were almost equal (the scale ranged from 1 to 5), and the difference was not significant (t = 0.217, $p = 0.82$).

For both music conditions (motivational music, non-motivational music), we conducted a two-way mixed ANOVA in which the music condition and liking of music were predictors, the evaluated environment was the within-subjects repeated factor, and the mean scores of particular evaluated dimensions were the dependent variables. We found a significant effect of music liking for the dimension *Mystery* and for the dimension *Pleasant*. For the dependent variable *Mystery,* an ANOVA revealed a significant effect of the environment ($F_{4,200} = 9.408$, $p = 0.001$, $\eta2 = 0.158$) and music liking ($F_{4,50} = 3.499$, $p = 0.014$, $\eta2 = 0.219$), while the effect of the musical condition was not significant ($F_{1,50} = 0.879$, $p = 0.363$, $\eta2 = 0.017$). For the dependent variable *Pleasant,* the ANOVA revealed a significant effect of the environment ($F_{4,200} = 23.889$, $p = 0.001$, $\eta2 = 0.323$) and music liking ($F_{4,50} = 3.175$, $p = 0.021$, $\eta2 = 0.202$), while the effect of the musical condition was not significant ($F_{1,50} = 0.25$, $p = 0.875$, $\eta2 = 0.001$).

It is also worth noting that an ANOVA conducted for the dimension *Interesting* revealed a tendency for music liking to influence the evaluation of the environment, although *p* lies slightly above the conventionally considered threshold for statistical significance ($p = 0.087$). For the dependent variable, an ANOVA revealed a significant effect of the environment ($F_{4,200} = 22.495$, $p = 0.001$, $\eta2 = 0.310$) and a nonsignificant effect of music

liking ($F_{4,50} = 2.164$, $p = 0.087$, η2 = 0.147), while the effect of the condition was not significant ($F_{1,50} = 0.060$, $p = 0.807$, η2 = 0.001). The ANOVAs conducted for the further spatio-cognitive emotional dimensions did not show a significant effect of music liking.

It is worth remembering that musical stimuli used in this study were chosen to be liked by participants from our previous study (*Franěk, van Noorden & Režný, 2014*). Clearly, not all participants in the present study shared their musical preferences with the participants, who originally selected these pieces, thus some differences in liking within the sample occurred. The results showed that the liking of specific music can be a factor that, to some extent, mediates the effect of music on the evaluation of environments in terms of certain spatio-cognitive and emotional dimensions, specifically, *Mystery* and *Pleasantness*. If people liked the music more, they evaluated the environments as more mysterious and more pleasant. There were no differences between the effects of both types of music

### Effects of disturbance of music

Finally, we analyzed responses to the item "The music disturbed me when evaluating the environment". The scores (the scale ranged from 1 to 5) for motivational music (Mean = 1.93, SD = 1.05) and non-motivational music (Mean = 1.59, SD = 1.26) differed, but not significantly ($t = 1.188$, $p = 0.26$). For both music conditions (motivational music, non-motivational music), we conducted a two-way mixed ANOVA where music condition and disturbance by music were predictors, the evaluated environment was the within-subjects repeated factor, and the mean scores of particular evaluated dimensions were the dependent variables. We found a significant effect of disturbance by music on *Coherence*. The ANOVA revealed a significant effect of the environment ($F_{4,200} = 16.579$, $p = 0.001$, η2 = 0.249) and a significant effect of disturbance by music ($F_{4,50} = 2.626$, $p = 0.045$, η2 = 0.174), while the effect of the condition was not significant ($F_{1,50} = 1.858$, $p = 0.8179$, η2 = 0.035). Furthermore, we found an almost significant effect ($p = 0.059$) of disturbance by music for *Mystery*. The ANOVA revealed a significant effect of the environment ($F_{4,200} = 9.219$, $p = 0.001$, η2 = 0.146) and a nonsignificant effect of disturbance by music ($F_{4,50} = 2.429$, $p = 0.059$, η2 = 0.163), while the effect of the condition was not significant ($F_{1,50} = 0.050$, $p = 0.825$, η2 = 0.001). The ANOVAs conducted for the further spatio-cognitive and emotional dimensions did not show significant effects of disturbance by music.

These results suggest that disturbance by music can be a further factor that mediates the effect of music on the evaluation of environments in terms of certain spatio-cognitive dimensions, specifically, *Coherence*. If people are disturbed by music, then they evaluate environments as less coherent, and they also tend to perceive the environment as less mysterious. There were no differences between the effects of both types of music.

## DISCUSSION

The goal of the present study was to investigate auditory and visual interactions and the effect of fast and slow music on the evaluation of outdoor urban built and urban natural environments. In contrast to our predictions, the present experiment failed to fully replicate the findings of the previous study conducted by *Yamasaki, Yamada & Laukka*
*(2015)*, which found that music had a pronounced effect, particularly on the activation rating of the evaluated environments. Our data revealed that there was only a slight and not significant influence of music on the evaluation of the environment. We found that the environmental features of the evaluated locations rather than music had a strong and significant effect on the evaluation of the environments.

To consider why we did not succeed in replicating the findings of *Yamasaki, Yamada & Laukka (2015)*, we first discuss the effect of the intensity of the music listened to while evaluating the environments. Although *Yamasaki, Yamada & Laukka (2015)* adjusted the sound level to an adaptive level for each participant, we may consider that for rock music, the participants adjusted the sound intensity to a higher level (*Fucci et al., 1993*). In contrast, in our experiment, the music was adjusted to a low level to create an auditory background where all musical information (melody, tempo, rhythm, timbre) was recognizable, but the music was not the center of attention. Although we did not manipulate the intensity of auditory stimuli in this experiment, we may speculate that the loudness of the music listened to while evaluating the environments is an important factor. Clearly, louder music captures more attention and thus may have a more pronounced effect on the evaluation of the environment.

However, further analysis of our data revealed that liking music was an important factor that mediated the effect of music in the evaluation of the environment, specifically in the dimensions *Pleasant* and *Mystery*. It seems that people like a particular type of music, they are more sensitive to its effect. Thus, the results also demonstrate the role of individual musical preferences. It is easy to understand that a positive feeling while listening to liked music can increase the perception of the pleasantness or interestingness of the environment. The explanation of the effect of liked music on the evaluation of the environment in terms of *Mystery* probably lies in the fact that this dimension describes a desire to further explore the environment; thus, it is combined with a positive emotional valence and with *Mehrabian & Russell*'s (*1974*) theory and their concept of *approach - avoidance behavior*. Approach behavior means that individuals tend to establish contact with the environment and stay inside it, while avoidance behavior means that they tend to avoid contact and to move away.

In strong contrast with *Yamasaki, Yamada & Laukka*'s (*2015*) study, we did not find any effect of music on perceived *activation*, even under exposure to fast motivational music that should motivate bodily movement. Although differences in the mean values for the perceived dimension of *Energy* were higher under exposure to music in both the less attractive environment 1 and in the calm environment 5 (see Fig. 1), the ANOVAs did not reveal significant effects. Moreover, we did not find any significant differences between the effects of motivational and non-motivational music, even though the selection of these types of music was based specifically on the effect of music to stimulate physical activity. Although it was found that the tempo of music while walking or running can have an unambiguous effect on movement speed (e.g., *Franěk, van Noorden & Režný, 2014*; *Laukka & Quick, 2013*; *Leman et al., 2013*; *Styns et al., 2007*), our results show that the same does not hold unequivocally for the perceived activation potential of the evaluated environment. Again, we should consider that the intensity of the music is likely an important factor.

Recently, *Yi & Kang (2019)* conducted research in public spaces of shopping malls and found that background music increases positive evaluation and approach behavior, while foreground music significantly increases arousal. It seems that the feeling of activation and energy may be better evoked by loud music than by a fast tempo itself.

In addition to the commonly used emotional categories *Activation* and *Valence* in research studying the effect of music on environmental perception, our study explored this effect in terms of spatio-cognitive dimensions used in environmental psychology to describe features of the preferred environment. However, we did not find an effect, except in the previously described spatio-cognitive dimension *Mystery,* which is associated with a positive valence in the evaluation of an environment. We can consider that in the perception of the *Coherence* of an environment, perceived congruence between music and the environment (*Yamasaki, Yamada & Laukka, 2015*) may play some role. It is possible that a high perception of congruence between music and the environment may increase the perception of coherence among particular features of the environment. In this experiment, we did not directly measure perceived coherence between visual and acoustical stimuli, but it was found that music that disturbed participants while evaluating the environment significantly decreased perceived coherence. Furthermore, we can speculate that more complex music, for instance, some genres of classical music (e.g., baroque counterpoint) that are perceived as complex and sophisticated (*Rentfrow & Gosling, 2003*), might eventually increase the perception of the *Complexity* of the environment, particularly in environments that are less complex. However, both types of music used in this experiment belonged to the category of "easy listening music". Thus, future research could also explore differences between the effects of music with a simple musical structure and a more complex one.

Although the aim of this study was to investigate audio-visual interactions and the influence of music on the evaluation of outdoor environments, our analysis revealed the highly significant effect of environmental properties on environmental evaluation. In general, we found that the urban natural environment is perceived as more pleasant, interesting, coherent, and mysterious than urban built environments regardless of the music heard. These findings are fully in accordance with a large body of environmental psychology research (e.g., *Herzog, 1989*; *Kaplan & Kaplan, 1989*; *Purcell & Lamb, 1984*; *Ulrich, 1981*). As in many environmental psychology studies, we used nice and attractive urban natural environments as visual stimuli. A question remains regarding how listening to music may affect perceptions of unkempt and unattractive natural settings.

In this study, we investigated the unilateral effect of music on the evaluation of an environment; however, whether one's liking of an environment might influence one's liking of the music listened to in this environment remains a question. We speculate that there may also be an influence of the environment on music liking, specifically, that music that is listened to in a well-liked environment is liked more than music listened to in a disliked environment. Unfortunately, in this experiment, we asked about participants' liking of music at the end of the entire experimental session, and we did not have data for their liking of music in the particular environments that were evaluated, which could have shed more light on this question. To shed light on the bilateral effect of music and

the environment, *Ehret et al. (2019)* investigated the influence of both spatial and musical surroundings on the perception of the overall atmosphere in two university rooms. They systematically manipulated the combination of two spatial surroundings and two pieces of music, with both pairs strongly differing in their valence. They found that musical and spatial valence both affected the overall experienced valence of the atmosphere.

There are also individual variables that may influence the effect of music on the evaluation of the environment. Specifically, *Chamorro-Premuzic & Furnham (2007)* reported that there are three different major uses of music. For some people, music serves mainly for emotional regulation and mood manipulation. A cognitive approach is typical for other individuals, which involves the rational or intellectual processing of music. Finally, background use of music is typical for people who use music as a background for social events or work. Different approaches to listening to music are, to some extent, related to levels of music education and activity. These factors might potentially influence the effect of acoustic information and the evaluation of the visual environment, which could be studied in future research. Furthermore, musically trained adults discriminate auditory differences more precisely than non-musically trained adults and are slightly better at sustained auditory attention than non-musically trained adults (e.g., *Carey et al., 2015*). Musically trained adults are also better than non-musically trained adults at preattentively extracting information from musically relevant stimuli (*Koelsch, Schröger & Tervaniemi, 1999*). Thus, in our task, musically trained adults may have been more influenced by music while evaluating the visual environment than musically non-trained ones. However, in the study by *Yamasaki, Yamada & Laukka (2015)*, the musical background of the participants was not described. Thus, music training is a variable to be included in future investigations.

Other factors responsible for differences between the findings of *Yamasaki, Yamada & Laukka (2015)* and the findings of our study may be cultural differences between the two samples. While in the former study the participants were female Japanese university students, the participants in our study were both men and women from the Czech Republic. The urban and natural environments differ substantially between these countries.

We may also consider differences in the research methodology between the pioneering study in this field and our study. In *Yamasaki, Yamada & Laukka*'s (*2015*) study, the participants listened to music only when they evaluated the environment; in our study, they listened to music during the entire experimental session, including the time when they were moving between the locations where they evaluated the environment. Therefore, it is possible that in the previous study, music had a greater effect on the evaluation of the environment because the participants listened to it only for a limited time with many interruptions. In this regard, our methodology created a setting more closely resembling a realistic situation in which people listen to music during their entire walking time without interruptions. On the other hand, people who use portable music players in outdoor settings decide to do so by themselves, and thus, their attitudes toward listening to music may be more intentional, and the intensity of their music might not be as low as in the present study. Thus, our experimental situation in some respect more closely resembled the scenario of listening to in-store background music, where people cannot select their

favorite pieces of music at their preferred intensity, rather than a scenario of the personal use of music in outdoor settings.

Our study has several limitations. First, for practical reasons we used a very brief online questionnaire that included only a single questions per dimension. Although we tried to choose a statement that would best capture the given dimension, this simplification may represent a limitation. Second, the study was conducted at the beginning of November. Clearly, the season, the form of vegetation, and even specific atmospheric conditions could influence audio-visual interactions and evaluations of an outdoor environment. A further limitation is the selection of music. Many variables might play a role. The musical style or genre as well as the timbre and selection of musical instruments and differences between vocal and instrumental music may play a role. The selection of the evaluated environments from the city of Hradec Králové could also be a further limitation. The results may differ in other urban environments (e.g., downtown areas of large cities) or in various types of natural environments (e.g., forests, mountains). Finally, in the discussion of our findings, we stressed the intensity of music as an important factor, but in this experiment, we did not manipulate this variable.

## CONCLUSION

The present study investigated the effects of audio-visual interactions on the perception of outdoor urban environments. In contrast with previous investigations (*Yamasaki, Yamada & Laukka, 2015*), we did not confirm the strong influence of music on the evaluation of an environment, but we showed that music increased positive evaluations in terms of certain spatio-cognitive and emotional dimensions of the environments (i.e., if the music was liked and not disturbing). We discussed diverse factors that may influence the results and particularly stressed the intensity of music given the difference between the effects of foreground and background music. These findings have some practical implications. Since some people combine the relaxing effect of outdoor walking with listening to music, it is useful to know how these acoustic and visual stimuli interact.

## ACKNOWLEDGEMENTS

We thank Michaela Jirková, Tomáš Vlček, Václav Kapler and Lukáš Charvát for their help in organizing and conducting the experiments.

### Funding

This work was supported by the Student Specific Research Grant 1/2020 from the Faculty of Informatics and Management at the University of Hradec Králové. The funders had no role in study design, data collection and analysis, decision to publish, or preparation of the manuscript.

## Grant Disclosures

The following grant information was disclosed by the authors:
Faculty of Informatics and Management at the University of Hradec Králové: 1/2020.

## Competing Interests

The authors declare there are no competing interests.

## Author Contributions

- Marek Franěk conceived and designed the experiments, performed the experiments, analyzed the data, prepared figures and/or tables, authored or reviewed drafts of the paper, and approved the final draft.
- Lukáš Režný conceived and designed the experiments, performed the experiments, prepared figures and/or tables, authored or reviewed drafts of the paper, and approved the final draft.
- Denis Šefara analyzed the data, prepared figures and/or tables, and approved the final draft.

## Human Ethics

The following information was supplied relating to ethical approvals (i.e., approving body and any reference numbers):

Committee for Research Ethics at the University of Hradec Králové approved the study (Ethical approval No. 8/2019).

## Data Availability

The raw data are available as a Supplementary Files.

## Supplemental Information

Supplemental information for this article can be found online at http://dx.doi.org/10.7717/peerj.9770#supplemental-information.

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
