# Peer review of "The effect of music on the perception of outdoor urban environment"

_PeerJ, doi:10.7717/peerj.9770_

## Round 0.1 · original submission · Major Revisions

Your paper looks like it will be publishable with revisions.

·

Basic reporting

This is generally a very clearly written article with a high level of professional English and it gives a good account of the background literature, follows the conventions of experimental reporting, provides clear overviews of all the relevant sections, and stands alone. The data has been shared appropriately.

I just have a few minor presentational points:

In the abstract I wonder whether you could rephrase the sentence about the key findings: ‘only very poorly and nonsignificantly’ could be better framed, perhaps, to say that there is only a slight and not significant influence of music on the evaluation of the environment.

I would suggest using the more precise acoustical terms of loudness or intensity (the latter is used in a few places) rather than ‘volume’ throughout.

A small point: Heye & Lamont (2010) found in their own results that participants often found the music enhanced rather than distracted from their perceptions of their surrounding environment, but this paper is cited in the introduction only in support of the auditory bubble hypothesis of Bull.

There are a few papers in this general area I am aware of which were not cited and the authors may wish to consider whether these are relevant: since there has been so little published on this specific topic, it might be useful to include these, and to take a further look at whether there are other relevant studies in this broader context.

Riener, C.R., Stefanucci, J. K., Profitt, D. R., & Clore, G. (2011). An effect of mood on the perception of geographical slant. Cognition and Emotion, 25(1), 174-182 (shows that manipulating mood through different types of music affects perception of slant in hill)
Bhattacharya, J., & Lindsen, J. P. (2016). Music for a brighter world: Brightness judgment bias by musical emotion. PLOS One, doi.org/10.1371/journal.pone.0148959. (experimental manipulation through music, happy music led to brighter, sad music led to darker perceptions of a grey square)
Ehret, S., Schroeder, C., Bernet, J., Holzmüller, & Thomaschke, R. (2019). All or nothing: The interaction of musical and spatial atmosphere, Psychology of Music, doi:10.1177/0305735619880288 (also experimentally investigates the effects of valence in music and spatial atmosphere in more controlled conditions).

Methods: line 214 should be ‘ninety-four’ not ‘nine-four’.

I would reframe the way the music stimuli are described as rather than being ‘borrowed’ from an earlier study they simply are used from it – and this is extremely good practice in the field, so should be celebrated (the use of ‘borrowed’ sounds a little apologetic). I wonder whether it would be better to label the types of music as ‘motivational’ and ‘non-motivational’ throughout, rather than the shorthand of ‘fast’ and ‘slow’ (e.g. line 235) or the variable names of Music1 and Music2 (which the reader doesn’t necessarily remember throughout – this is used in the figures). I think it would also be useful to emphasise here, given the importance of liking later on, that this music was chosen to be liked by the participants, and to remind the reader of this in the liking section of the results.

Experimental design

The study includes a well defined research question and a highly appropriate range of experimental variables (type of music – fast/slow; type of environment – urban build/urban natural; outcome measures (emotional and spatio-cognitive dimensions), with control applied to the intensity dimension to make the music only background.

It would be helpful perhaps to show some photographs of the environments used, which might help in relation to the comparison with the previous study and also to elaborate what ‘openness’ means (lines 283/4) as this was not entirely clear. If not, then a fuller verbal description might be useful, particularly in Table 2.

The ratings are quite brief (single questions per dimension) compared to the Yamasaki et al. study, and while this was obviously helpful from a practical level I would like to see some evaluation of the effects this might have had on responses.

Validity of the findings

Overall the results are reported appropriately and the findings are generally clear, with appropriate interpretations drawn (with one exception noted below about music experience/training).

Liking for the music is mentioned in the results but there could be more discussion about liking for the environments. Presumably, given the importance of the emotional responses to the environmental variables, this will also play a role, and even though explicit data was not gathered, there could be some speculation about this effect (the pleasantness dimension might be something to pull out even more). The assumption is that the music influences the environment but it is also possible that the relationship goes the other way, which I think is what was intended by the analysis of music liking as a mediator and this point could be brought out more explicitly.

In terms of speculation, the points about musical experience in the Discussion should, in my view, be toned down and reframed somewhat. I would like to see ‘musicians’ reworded as ‘musically trained adults’ because arguably everyone is a musician. More substantially, there is some speculation here about the musical backgrounds of participants without appropriate support from any systematically gathered data and I think the point made ought to be toned down to suggest that music training is a variable to be included in future investigations. I would also suggest that this could be broadened out to include a measurement of active engagement with music, such as the Goldsmiths Musical Sophistication Index Active Engagement scores which provide information about the amount of music listened to, interest in music listening and so on which might help explain individual differences in results.

Additional comments

Overall this strong paper contributes to a small but important field, of relevance to the journal, and taking account of the above relatively minor comments this would make a valuable addition to the literature.

Reviewer 2 ·

Basic reporting

1. This study examined whether background music can modify the impression of a visual environment. The article is written in clear English.

Experimental design

The experimental design is good.

Validity of the findings

No comments.

Reviewer 3 ·

Basic reporting

I think that this article is written by professional English in general. All sentences are clear and unambiguous. But, there are small typos between “,” and “.” in p-values in line 396 and 399.

The references cited in the introduction are adequate and new. However, the list of references should be corrected because the order, especially from the initial “k” to “l”, is confused.

Experimental design

Research question is clear and within the scope of the journal.

In this experiment, the characteristics of musical materials are very important. Although authors focus on the motivational characteristics of music listened to, the musical characteristics which affect the evaluation of given environments are not limited to it. Therefore, more detailed descriptions of the musical materials are desirable, such as semantic profile of the musical materials by SD method. If it is confirmed that Music 1 and Music 2 differed extensively in many aspects of those perceived characteristics, authors’ conclusion, the poor influence of music on the evaluation of environments, will be more persuasive.

Validity of the findings

It is really important findings that the liking of music and the disturbance of music mediate the effect of music. But, according to the descriptions, authors drew this conclusion from ANOVAs in which the liking score was dependent variable with five levels. If so, multiple comparison tests are necessary for the investigation of the direction of effects of music liking. About the disturbance of music, the same can be said.

In this study, participants heard music at lower level than a previous study by Yamasaki et al. (2015). They heard music not only at each selected environment, but also during moving between environments. Authors consider these experimental conditions as more realistic than Yamasaki et al.. Indeed, continuing to hear music during moving may be closer to the everyday music listening situation of participants. However, because listeners with portable music players in outdoor settings decide to do it by themselves, their attitude to listening may be more intentional and their volume of music will be not so low. In addition, it seems to be less realistic that participants hear only a song during a lasting event. Thus, some features of the experimental conditions which authors set up seem to be closer to ones of in-store settings with background music rather than ones of personal use of music in outdoor settings. This doesn’t reduce the value of findings in this study, but influences the theoretical and the practical meanings of findings. Therefore, more discussion on this topic is desirable.

Additional comments

I believe that the present study is worth publishing. I hope that my suggestions are helpful to your work.

In order to understand the results of this experiment, it is very helpful for readers to grasp the experimental stimuli clearly. Music stimuli can be listened to easily from Youtube etc., while visual environments are difficult to be seen. So, pictures will be very helpful, if possible.

---

## Round 0.2 · accepted · Accept

Thank you for making the changes suggested by the reviewers.